# Which Is the Best Way to Treat Massive Hemoptysis? A Systematic Review and Meta-Analysis of Observational Studies

**DOI:** 10.3390/jpm13121649

**Published:** 2023-11-26

**Authors:** Eleni Karlafti, Dimitrios Tsavdaris, Evangelia Kotzakioulafi, Leonidas Kougias, Georgios Tagarakis, Georgia Kaiafa, Smaro Netta, Christos Savopoulos, Antonios Michalopoulos, Daniel Paramythiotis

**Affiliations:** 1Emergency Department, University General Hospital of Thessaloniki AHEPA, Aristotle University of Thessaloniki, 54636 Thessaloniki, Greece; 21st Propaedeutic Department of Internal Medicine, University General Hospital of Thessaloniki AHEPA, Aristotle University of Thessaloniki, 54636 Thessaloniki, Greece; ekotzaki@auth.gr (E.K.); gdkaiafa@auth.gr (G.K.); csavvopo@auth.gr (C.S.); 31st Propaedeutic Surgery Department, University General Hospital of Thessaloniki AHEPA, Aristotle University of Thessaloniki, 54636 Thessaloniki, Greece; tsavdaris@auth.gr (D.T.); smaronetta2@gmail.com (S.N.); amichal@auth.gr (A.M.); danosprx@auth.gr (D.P.); 4Department of Radiology, University General Hospital of Thessaloniki AHEPA, Aristotle University of Thessaloniki, 54636 Thessaloniki, Greece; leonidaskougias@gmail.com; 5Department of Cardiothoracic Surgery, University General Hospital of Thessaloniki AHEPA, Aristotle University of Thessaloniki, 54636 Thessaloniki, Greece; gtagarakis@auth.gr

**Keywords:** massive hemoptysis, management, bronchial artery embolization, systematic review, meta-analysis

## Abstract

Introduction: Hemoptysis is one of the most common symptoms of respiratory system diseases. Common causes include bronchiectasis, tumors, tuberculosis, aspergilloma, and cystic fibrosis. The severity of hemoptysis varies from mild to moderate to massive hemoptysis and can easily lead to hemodynamic instability and death from suffocation or shock. Nevertheless, the most threatening hemoptysis that is presented to the emergency department and requires hospitalization is the massive one. In these cases, today, the most common way to manage hemoptysis is bronchial artery embolization (BAE). Methods: A systematic literature search was conducted in PubMed and Scopus from January 2017 (with the aim of selecting the newest possible reports in the literature) until May 2023 for studies reporting massive hemoptysis. All studies that included technical and clinical success rates of hemoptysis management, as well as rebleeding and mortality rates, were included. A proportional meta-analysis was conducted using a random-effects model. Results: Of the 30 studies included in this systematic review, 26 used bronchial artery embolization as a means of treating hemoptysis, with very high levels of both technical and clinical success (greater than 73.7% and 84.2%, respectively). However, in cases where it was not possible to use bronchial artery embolization, alternative methods were used, such as dual-vessel intervention (80% technical success rate and 66.7% clinical success rate), customized endobronchial silicone blockers (92.3% technical success rate and 92.3% clinical success rate), antifibrinolytic agents (50% clinical success rate), and percutaneous transthoracic embolization (93.1% technical success rate and 88.9% clinical success rate), which all had high success rates apart from antifibrinolytic agents. Of the 2467 patients included in these studies, 341 experienced rebleeding during the follow-up period, while 354 other complications occurred, including chest discomfort, fever, dysphagia, and paresis. A total of 89 patients died after an episode of massive hemoptysis or during the follow-up period. The results of the meta-analysis showed a pooled technical success of bronchial artery embolization equal to 97.22% and a pooled clinical success equal to 92.46%. The pooled recurrence was calculated to be 21.46%, while the mortality was 3.5%. These results confirm the ability of bronchial artery embolization in the treatment of massive hemoptysis but also emphasize the high rate of recurrence following the intervention, as well as the risk of death. Conclusion: In conclusion, massive hemoptysis can be treated with great clinical and technical success using bronchial artery embolization, reducing mortality. Mortality has now been reduced to a small percentage of cases.

## 1. Introduction

Respiratory diseases are among the most common reasons for patients visiting hospitals and emergency rooms. These diseases are most commonly manifested by one or more symptoms: cough, chest pain, shortness of breath, or hemoptysis [1]. Further, the SARS-CoV-2 virus infection that broke out recently and quickly spread in a pandemic appears with these symptoms (most commonly, cough and shortness of breath) [2,3]. Hemoptysis may therefore not be the most common clinical finding in COVID-19 patients [4], but it is an urgent situation that requires observation, finding the cause, and taking steps to prevent a more severe recurrence.

Hemoptysis is defined as the expulsion of blood through the oral tract from a source located in the lower respiratory tract, i.e., below the level of the glottis [5,6]. It is a frequent clinical finding that is usually due to bronchitis, pneumonia, bronchiectasis, tuberculosis, neoplasms (bronchial carcinoma, intrabronchial tumors, sarcomas, and metastases), or even pulmonary embolism [5,6,7,8,9,10]. Other less common causes include coagulopathy, platelet disorders, pulmonary hypertension, aneurysm rupture, trauma (catheter, biopsy, iatrogenic, or blunt trauma), foreign body aspiration, vascular and collagen diseases (such as Goodpasture’s syndrome, Behçet’s disease, systemic lupus erythematosus, granulomatosis with polyangiitis, Henoch–Schonlein purpura, and rheumatoid arthritis), drugs and narcotics (such as anticoagulants, penicillamine, solvents, crack, and cocaine), and even gastric content aspiration or cryptogenic causes [5,6,7,8,9,10,11,12]. Its extent can vary from the presence of blood in the sputum to potentially life-threatening massive hemoptysis [8]. The etiology of hemoptysis affects the choice of appropriate treatment [5,6,9,11].

Massive hemoptysis is defined as blood loss that exceeds 600 mL per twenty-four hours or 150 mL per hour [8]. Massive hemoptysis is related to pathologies that alter the pulmonary vasculature, usually affecting the bronchial arteries, and can be either inflammatory or neoplastic. This emergency can easily and quickly lead to death due to airway obstruction and hypoxemia from asphyxiation, rather than hemodynamic disturbance and shock, which do not occur early [9]. Even a tiny vessel rupture can have significant clinical consequences because of aspiration and the difficulty of stable thrombus formation that prevents vessel healing in thin-walled airways. Due to the severity of the situation, it is necessary to differentiate between the conditions with a similar picture which are characterized as pseudohemoptysis [13]. These can be upper digestive, upper respiratory, or even oral cavity bleeding. Hemoptysis has a bright red color as well as an alkaline frothy sputum, while in other cases of pseudohemoptysis the blood is acidic and darker in color [10,13].

Two pathways are blamed as pathophysiological mechanisms. The first and most common one concerns bleeding from the bronchial arteries because they handle higher pressures [10,14,15,16,17]. This pathway begins when diseases cause hypoxic vasospasm and thereby limit pulmonary circulation. This implies an increase in blood supply with an increase in the rupture of the fragile anastomotic vessels and bleeding into the alveoli and bronchi. Other causative diseases are those that, through the release of angiogenic growth factors, cause neovascularization and the development of collateral circulation. These vessels are also fragile, and so their rupture causes hemoptysis. Hemoptysis, however, can also occur from the vessels of the pulmonary circulation, either iatrogenically during catheterization, during aneurysm rupture, or from other causes [10,14,15,16,17].

According to the guidelines [5,9,10,11,18], the first step is to diagnose massive hemoptysis. The diagnostic algorithm includes a general blood test as well as a coagulation profile. The imaging examination usually does not start with an X-ray because the diagnostic accuracy and detection of bleeding are poor. For this reason, other imaging tests are preferred. These include bronchoscopy, which can detect bleeding in over 90% of cases. Another advantage of this examination is that it does not require moving the patient, while at the same time it offers information about the patient’s anatomy, a fact that facilitates subsequent intervention, increasing its effectiveness [19,20,21]. An alternative method is computed tomography (CT), which has been shown to be even more effective in detecting bleeding and can therefore replace bronchoscopy. Although it is an unusual finding, bronchial artery bleeding can be seen at the time of the scan. This can indicate the site and cause of the bleeding. Native CT can also indicate the site of bleeding, which may manifest as a ground glass infiltration or as pulmonary consolidation, and permit the identification of the site of major architectural distortion in the case of diffuse pathologies [22]. CT angiography permits the identification of the origin of bronchial arteries from the thoracic aorta and facilitates catheterization during interventional angiography. A bronchial artery diameter greater than 3 mm is usually considered pathologic. There is great variability in the number and site of origin of the bronchial arteries, but the most usual patterns are one or two arteries for each lung that originate from the descending thoracic aorta around the level of the tracheal bifurcation. Sometimes, a bronchial artery may arise as a common trunk with an intercostal artery. On the other hand, lung pathology located superficially may result in neovascularization from intercostal artery branches or from internal mammary arteries [5,9,10,16,23,24,25].

Then, according to the guidelines, the hemoptysis is treated [26,27,28]. The treatment of massive hemoptysis has two aspects. The first concerns the emergency arrest of bleeding which does not stop spontaneously and constitutes an emergency. This can be accomplished by using a rigid bronchoscope and adequate aspiration of blood, as well as intubation of the lung and occlusion of the bleeding site, e.g., with a balloon, to ensure a non-bleeding lung. Finally, hemostatic drugs, such as pituitrin, hemocoagulase, and others such as carbazochrome sulfonate sodium and carbazochrome tablets, are applied. Therefore, after the diagnosis is made and emergency treatment is applied, where necessary, follow-up targeted treatment of massive hemoptysis is required. In this phase, the main technique used is bronchial artery embolization (BAE). Alternatives are surgical treatment of the bleeding or etiological treatment [5,9,10,11,18]. The prognosis of massive hemoptysis is indicated by several factors. These include the amount of blood in the sputum, the etiology of hemoptysis (such as cancer), and hemodynamic instability [29,30].

Furthermore, for the treatment of massive hemoptysis, it is important to take certain factors into serious consideration. These include coagulation parameters as well as the taking of anticoagulant drugs. Specifically, the number of platelets determines the safety of both biopsy (safety thresholds of more than 50,000/mm^3^) and bronchoscopy (safety thresholds of more than 20,000/mm^3^) [31,32]. Anticoagulation, especially colpidogrel, also increases the risk of bleeding during procedures such as biopsies [33]. From the laboratory findings, urea and BUN are also considered important, as uremia and BUN > 30 mg/dL increase the risk of bleeding during surgery [34].

Since massive hemoptysis is a life-threatening condition, an effective treatment method is essential. Massive hemoptysis, unlike mild hemoptysis, is primarily treated invasively. BAE is considered the method of choice and the first in line. This method can be applied with various techniques. One of the most common is the Seldinger technique. The Seldinger technique usually uses the right femoral artery for access and selective catheterization of the bronchial and intercostal arteries using a 5 French catheter (Cobra, Simmons or DESLER), which is inserted through the diagnostic catheter a few centimeters distally to the vessel origin. This setup provides stability and impedes reflux of embolic material to the thoracic aorta or into radiculomedullary vessels, which, when present, arise as proximal branches [35,36,37]. Catheterization of bronchial arteries and contrast injection will demonstrate enlarged, tortuous vessels with intense angiographic blush in the parenchymal phase of the angiography. This pathologic imaging pattern is enough to justify embolization, given that active contrast extravasation is usually not observed. Each angiographic run should also be thoroughly examined to identify possible branches directed to the spinal canal. Sometimes, radiculomedullary arteries may arise from bronchial arteries as anatomic variations or from a costobronchial trunk. The dominant radiculomedullary artery is the artery of Adamkievitch and provides a major contribution to the anterior spinal artery. This branch presents a characteristic hairpin configuration and projects on the vertebral column along the midline. If angiographic interpretation is difficult, rotational angiography with 3D reconstructions might help. Superselective catheterization of anomalous arteries using 2-F microcatheters is chosen, where possible, for the best results. Several different techniques for access and embolization may be used as appropriate. Also, embolic agents, such as polyvinyl alcohol, coils, gelatin sponges, and Embospheres, are more common [35,36,37]. The embolic materials of choice are particles or microspheres. They should not be smaller than 250 microns, otherwise the probability of their entering dangerous collaterals, leading to non-target embolization, is increased. Coils should be avoided because a new embolization session in case of a relapse will be impossible [17]. It is important to identify and embolize the bronchial artery that supplies the responsible pulmonary lobe, but in cases of diffuse pathologies, embolization of all identified pathologic vessels can be considered, case by case, to prevent future bleeding episodes. The double vascularization of the pulmonary parenchyma usually prevents ischemic complications.

Therefore, knowledge regarding the treatment of massive hemoptysis is necessary to reduce mortality risk. This systematic review aims to evaluate the available treatments for massive hemoptysis according to the latest available studies in the literature, including evaluation of the technical and clinical success of the methods for managing hemoptysis, as well as recurrence, complications, and mortality. The technical success of an operation is defined as successful intervention in massive hemoptysis, i.e., successful bronchial embolization of the bleeding arteries and arrest of the bleeding. Clinical success is defined as the resolution of the clinical findings of massive hemoptysis after the application of the intervention, that is, the cessation of hemoptysis and other clinical findings that the patient may present.

## 2. Materials and Methods

### 2.1. Study Protocol and Guidelines

This systematic review was written according to the Preferred Reporting Items for Systematic Reviews and Meta-Analyses (PRISMA) guidelines [38]. This systematic review is registered in the Open Science Framework (OSF) with the registration number: DOI osf.io/r2n8w.

### 2.2. Eligibility Criteria

The eligibility criteria for this systematic review are presented and were defined with the PICO framework. To be included in this systematic review, studies had to meet the following inclusion criteria:Involve adult patients with massive hemoptysis;Report on the management of massive hemoptysis;Provide technical and clinical success rates of the treatment used;Include data on recurrence and mortality.

Reviews, letters, comments, and case reports were excluded. Only original articles that met the above criteria were included.

### 2.3. Information Sources, Search Strategy, and Selection Process

A systematic literature search was conducted in two databases (PubMed and Scopus) using the keywords ‘massive hemoptysis’ and ‘management’. The search was limited to a six-year period from January 2017 (with the aim of selecting the newest possible studies in the literature) until May 2023 and articles had to be written in English to be selected. The selection process was carried out by two independent reviewers (D.T. and E. Kar.) using Rayyan [39], who assessed the studies for inclusion in this systematic review. The two reviewers were blinded during both the title and abstract screening and the full-text screening. Any disagreement was resolved by a third reviewer (E. Kot.).

### 2.4. Data Collection Process and Data Items

Data extraction was carried out by one reviewer (D.T.), and another reviewer (E.Kot.) independently checked the results. The data were extracted in a standardized Excel form. Data extraction was performed without the use of any automation tools. From the studies that were reviewed, the relevant data for extraction were: the main characteristics of each study; the identity of each study (author and year of publication); the total number of participants, along with the mean age and the percentage of men; the type of study; the management of bleeding; and the technical success rate of the method used.

### 2.5. Quality Assessment

Quality assessment of the included studies was performed using the Newcastle–Ottawa Quality Assessment Scale for Cohort Studies by two independent reviewers (D.T. and E.Kar.). Any disagreement was resolved by a third reviewer (E.Kot.). The Newcastle–Ottawa tool consists of three evaluation domains. These are the selection of the participating patients, comparability, and outcomes. The selection of participants includes: the representativeness of the exposed cohort, the selection of the non-exposed cohort, the ascertainment of exposure, and a demonstration that the outcome of interest was not present at the start of the study. With respect to the first question regarding the representativeness of the exposed cohort, a star is given when the sample is truly or somewhat representative. Regarding the selection of the non-exposed cohort, a star is given if the selection of the non-exposed cohort is drawn from the same community as the exposed cohort. To be given a star for the ascertainment of exposure, a secure record or structured interviews must be presented. Finally, the last star is given if there is a demonstration that the outcome of interest was not present at the start of the study. Regarding the 2nd domain, on comparability, two stars may be given for the comparability of cohorts if the design or analysis controlled for confounders. For the outcomes, the assessment of outcomes and the follow-ups are evaluated. For the first, a star is given if the assessment was independent and blind or a record linkage, while for the follow-up, a star is given if the assessment was sufficient to evaluate the results and if several of the participants were included in it.

### 2.6. Statistical Analysis

Statistical analysis was conducted with R studio (R Core Team (2022). R: A language and environment for statistical computing. R Foundation for Statistical Computing, Vienna, Austria. URL: https://www.R-project.org/ (accessed on 8 September 2023)). A proportional meta-analysis was conducted using the metafor [40] and meta packages [41] using a random-effects model. Cases and total numbers of participants were extracted for each outcome from each individual study, and their proportions were calculated during the meta-analysis. Publication bias was determined with visual inspection of funnel plots for each outcome and Egger’s test [42]. Heterogeneity is presented with I^2^ results, and results between 0 and 40% were considered instances of non-important heterogeneity, those between 40 and 60% instances of moderate heterogeneity, those between 60 and 75% instances of substantial heterogeneity, and those between 75 and 100% instances of considerable heterogeneity [43,44].

## 3. Results

### 3.1. Study Selection

A total of 785 studies were found using the keywords ‘treatment’ and ‘massive hemoptysis’, 395 of them in PubMed and 390 in Scopus. Of these, 658 were excluded, as they were either duplicates or letters, case reports, reviews, or comments. Thus, the remaining 127 studies were screened for eligibility, 30 of which met the inclusion criteria and were assessed for eligibility in the full-text screening. These studies therefore analyzed the success rate of the management of massive hemoptysis. The screening process for the studies selected for inclusion in this systematic review is illustrated in a PRISMA 2020 flow chart (Figure 1). All selection stages from the initial stage to the final stage are depicted.

### 3.2. Study Characteristics

#### 3.2.1. Major Characteristics of the Included Studies

The 30 studies included 2647 patients who developed massive hemoptysis and were treated accordingly [45,46,47,48,49,50,51,52,53,54,55,56,57,58,59,60,61,62,63,64,65,66,67,68,69,70,71,72,73,74]. In these studies, treatment was applied either to stop bleeding in cases of massive hemoptysis in which conservative treatment was unsuccessful or to prevent recurrent bleeding. All these studies were retrospective observational cohort studies. The largest sample size was 489 patients, in Ishikawa et al., 2017 [49]. This study was conducted in Japan by a center specializing in hemoptysis. In contrast, the smallest sample size was found in Clements et al., 2022 [72], with only three patients. Finally, the majority of studies (26/30) used bronchial artery embolization for managing massive hemoptysis. Despite the common methods applied in all these investigations, the different tools, materials, and techniques applied in each investigation are of interest. These are presented in Table 1.

#### 3.2.2. Findings of the Studies

Some studies used techniques that differed greatly or slightly from bronchial artery embolization or used the classical technique of bronchial artery embolization with various variations depending on the circumstances and characteristics of the patients. Kucukay et al. [56] used large-sized (700–900 lm) tris-acryl microspheres (Embospheres) for bronchial embolization. The results showed the great clinical success of this technique, with very high long-term success rates. Specifically, clinical success reached 100%. The disadvantage of this technique is that the microspheres, due to their large size, can occlude the bronchial artery, which is much more central to the lesion. However, the use of a microcatheter significantly reduces the likelihood of this happening.

Xiaobing et al. and Seki et al. [59,61] used chemotherapeutic substances as means of embolization of the coronary arteries. This variant is called bronchial artery chemoembolization and seems to be advantageous over the classic technique with respect to the long-term results of treating hemoptysis due to lung cancer. It requires repeated courses of treatment, and various substances, such as epirubicin, nedaplatin, and etoposide, are used. The trophic arteries of the tumor are embolized, and it seems to be a technique that solves the problem of the difficult management of hemoptysis in cancer patients. This difficulty is due to the different and complex pathophysiology of hemoptysis in lung cancer. BAE, on the other hand, does not manage the tumor but only the hemoptysis, leading to the high recurrence rates in these patients.

In cases of anomalies, such as pseudoaneurysms, Marcelin et al. [63] suggested the use of pulmonary embolization versus bronchial embolization or endoprosthesis placement. This technique scored a high success rate relative to the difficulty and severity of the condition, while two patients died due to massive bleeding. The stent ensures the patency of the vessel; however, a complication is its occlusion due to the stent.

Mehta et al. [65] evaluated the use of customized endobronchial silicone blockers for the management of massive hemoptysis in cases where BAE is not an option. With this technique, a customized endobronchial silicone blocker is used to wedge and occlude the bleeding, with quite favourable results. The clinical success rate reached 92.3%, making this technique successful enough to replace BAE where it is not possible. Its only flaw is that it has not been studied as well as BAE to ensure its safety and success. Yang et al. [73] used the dual-vessel intervention (DVI) technique. This is a less successful technique that scored technical and clinical rates of 80% and 66.7%, respectively. This technique includes bronchial and pulmonary artery embolization with a hyperselective catheter and the use of PVA and a spring coil. Finally, the lowest clinical success rate was achieved in Samkari et al. [58], who used antifibrinolytic agents to manage hemoptysis. However, this percentage was also affected by the population group, as it included patients with cystic fibrosis, who have high failure rates in operations.

An et al. [47] examined bronchial artery embolization with a specific opening, namely, the opening of the lower wall of the aortic arch. This study reports that this method was particularly effective, with fewer complications and rebleeding, and involved the use of the JL4 catheter. The use of this catheter is particularly important, as operations of this type have not been as successful as operations using other catheters. The advantages of this catheter are found in its shape, its rigidity, and the absence of the need to create a loop.

Finally, Cheng et al. [45] analyzed the safety and effectiveness of the use of embospheres alone or in combination with gelfoam particles for BAE. The results showed that the combination of these gave significantly better results, with technical and clinical success scores of 97.14% and 97.14%, respectively. The use of pure embospheres scored technical and clinical success rates of 92.99% and 85.96%, respectively. Therefore, the major difference between these two is found in the clinical success of the two techniques. For this reason, the authors suggest the use of embospheres together with gelfoam particles to avoid clinical failure.

##### Etiology of Massive Hemoptysis

In the studies included in this systematic review, bronchiectasis is reported as the first cause, in 658 reported cases out of 2647 in total, thus covering a quarter of the causes of massive bleeding (Table 2 and Figure 2). This is followed by tumors and tuberculosis, comprising 444 and 397 cases, respectively. Cases of aspergilloma and cystic fibrosis are also common and require special management.

##### Radiological and Angiographic Findings

In the studies included in this systematic review, 460 cases of hypertrophy and dilatation, 44 cases of parenchymal hypervascularity, 145 cases of parenchymal blush, 73 cystic findings, 37 cavities, and 78 aneurysms were reported (Table 3). However, these numbers underestimate the actual findings because many of the studies did not report the number of findings included in the overall result.

##### Recurrence, Complications, and Mortality

Table 4 presents the main postoperative complications, rebleedings, and deaths that were recorded. The main complication was chest pain, sometimes mild and sometimes more severe, which required investigation and treatment with analgesics. At least 140 such cases were reported in the studies included in this systematic review. The next most-frequent complication was fever (52 cases reported), but no further details were available from the studies regarding its cause. Only 11 cases of dysphagia and throat discomfort were reported, while even fewer, i.e., 2, were cases of paresis, one of the serious complications of the operation. A total of 149 more complications were reported, including nausea/vomiting (the most common), headache, and abdominal pain, while some of the serious and life-threatening complications reported were aortic dissection, cerebellar infarctions, and hematomas.

Regarding the occurrence of rebleeding, 341 cases occurred out of 1979 cases, while it is not known what happened to the remaining 668 cases due to a lack of data. These cases of rebleeding, however, were recorded at different follow-up times, that is, some appeared in the first month after the operation, others in the first six months, others in the first year, and so on. A total of 89 deaths related to hemoptysis were also recorded, which highlights the seriousness and danger of the condition, as well as the need for proper and effective treatment.

### 3.3. Quality Assessment

The results from the quality assessment using the Newscastle–Ottawa Scale are listed in Table 5. The included studies did not use non-exposed cohorts and were not given a star in the selection domain.

### 3.4. Meta-Analysis Results

A proportional meta-analysis was performed for all studies that used a common mode of management of massive hemoptysis, that is, BAE, and provided relevant data [45,46,47,48,49,50,51,52,53,54,56,57,59,60,61,62,63,64,66,68,69,70,71,72,74].

#### 3.4.1. Technical Success Meta-Analysis

The results of the statistical analysis showed a pooled technical success rate of 97.22%, confirming the technical success of the operation. Technical success is defined as successful embolization of the bronchial arteries. These results are presented with a forest plot in Figure 3. Significant heterogeneity between the included studies was observed (I^2^ = 77%, *p* < 0.001). Publication bias for technical success presented significant asymmetry (Figure 4, *p* = 0.0024).

#### 3.4.2. Clinical Success 

Clinical success is defined as complete cessation of hemoptysis after bronchial arterial embolization. The statistical analysis of the results of 24 studies showed a pooled clinical success equal to 92.46% (90.43; 94.50 95% CI), with moderate heterogeneity (I^2^: 57%, *p* < 0.001). These results confirm the effectiveness of BAE in treating massive hemoptysis. No significant asymmetry was observed in the funnel plot regarding publication bias (*p* = 0.0772). The results are presented in Figure 5 and Figure 6.

#### 3.4.3. Recurrence

All episodes of recurrence and rebleeding recorded in the studies in this systematic review during follow-up were used to calculate the prevalence of recurrence. The pooled recurrence was calculated to be equal to 21.46% (14.04; 28.89 95% CI), which shows the frequent occurrence of recurrence in patients, which can often be life-threatening. The result was considered statistically significant, with *p* < 0.0001, while high heterogeneity was also distinguished. These results are visible in Figure 7 and Figure 8, below.

#### 3.4.4. Mortality

The pooled result for mortality was 3.5% (95% CI: 1.78; 5.21). Mortality, although it is a rare complication of massive hemoptysis, is a real risk that has not yet been eliminated. Significant asymmetry was observed in the funnel plot, indicating publication bias (*p* = 0.0002). The results are presented in Figure 9 and Figure 10.

## 4. Discussion

In this systematic review, 30 studies related to the management of massive hemoptysis were included, of which 26 used bronchial artery embolization as a management method [45,46,47,48,49,50,51,52,53,54,56,57,59,60,61,62,63,64,66,67,68,69,70,71,72,74]. The 30 studies were observational retrospective cohort studies. As a result, no definitive conclusions can be made about the success of the methods for treating massive hemoptysis. The technical success of BAE ranges from 73.7% to 100%, while the clinical success ranges from 82.1% to 100%. This large variation depends on various factors, such as the technique and agents used, as well as the cause of the hemoptysis. Regarding the techniques, these initially include access to the vascular tree of the pulmonary parenchyma. This usually occurs via the femoral artery, although the transaxillary route has also been used. Embolization of the visible abnormal arteries then follows. The choice of the appropriate embolization factors is also important, as it has been found that they affect the success of the operation [35,36,37,75,76]. The most common agent is polyvinyl alcohol; however, a combination of agents has been found to slightly increase the technical success but significantly increase the clinical success [35,36,37,75,76]. Cheng et al. [45] showed that with the combination of embospheres and gelfoam particles, the technical success increased by 4.18%, while the clinical success jumped from 85.96% (without gelfoam particles) to 97.14%. It is worth mentioning the value of superselective arterial catheterization, which achieves the best results with the fewest possible complications [35,36,37,75,76].

The results of the meta-analysis demonstrate the success of BAE in the treatment of massive hemoptysis. Specifically, BAE scored a pooled technical success equal to 97.22% as well as a pooled clinical success equal to 92.46%. Therefore, BAE can with great ability both achieve arterial embolization and interrupt massive hemoptysis. However, the level of recurrence seems to be high (21.46%). Therefore, despite the immediate success of the intervention, the long-term success is limited due to the high recurrence numbers. Finally, mortality is limited to 3.5%, which also confirms the success of the intervention.

Surgical treatment is an alternative to BAE, which, however, is not minimally invasive, instead requiring a long operation with lobectomy of the bleeding lung segment. It is chosen in patients in whom temporary hemostasis or BAE will not produce a long-term effect, leading to rebleeding in a short period of time. It is a prerequisite that the patient is cardiorespiratorily fit for such an operation and that the bleeding is unilateral. Etiological therapy is chosen for respiratory diseases that do not cause anatomical damage and must be accompanied by respiratory support by administering oxygen [10,11,18]. It is the method of choice for conditions such as aspergilloma as well as for iatrogenic injuries that cause massive hemoptysis [77,78].

In the current systematic review, we defined massive hemoptysis as the expulsion of blood through the oral route in an amount exceeding 200 mL per twenty-four hours. In smaller amounts, hemoptysis is categorized as mild or moderate. In these cases, the patient is usually hemodynamically stable and the way to deal with hemoptysis changes, including treatment of the etiology that causes hemoptysis, and thus, together with the treatment of the cause of the hemoptysis, the hemoptysis itself is removed. However, it has been found that hemoptysis can be treated in these cases with the use of nebulized tranexamic acids. These are synthetic analogues of lysine with anti-fibrinolytic action. Specifically, they cause the suspension activation of plasminogen to plasmin and block the action of plasmin on fibrin. In the double-blind, randomized study by Wand et al., they were found to score 96% in the treatment of hemoptysis versus placebo, which achieved only 50%. This confirms the success of TAs in the management of mild-to-moderate hemoptysis. Also, the study showed that co-administration of antibiotics with TA does not increase the success of the intervention [14,79,80,81,82].

However, it has been found that moderate hemoptysis can potentially have the same prognosis of rebleeding as massive hemoptysis as well as the same mortality. For this reason, it is suggested that it be treated in the same way as the massive one. In other words, it is recommended to use BAE to treat it rather than etiological treatment or TA. In terms of success rates, these show the significant ability of BAE to manage moderate hemoptysis, scoring more than 96% and 94% for technical and clinical success, respectively [14,79,80,81].

Many factors have been implicated as risk factors for hemoptysis in the population. Initially, it was found that bronchiectasis is associated with an increased risk of hemoptysis, as well as hypertrophic bronchial arteries that are associated with heavier and faster hemodynamic instability [6,9,16,17,83,84]. As for hypertrophic arteries, it has been found that these are located in patients carrying the mutated BMPR2 gene. However, this mutation is not associated with an increased risk of hemoptysis but only with an increased frequency of bronchial artery hypertrophy. Smoking also increases the risk of developing hemoptysis, while anticoagulants and antiplatelet drugs increase both the risk of hemoptysis and the severity of this hemoptysis. Diabetes was also implicated as a risk factor, perhaps due to the vascular damage it causes. Hemoptysis occurs more often in males, while it occurs less often in diseases with a course between 1 and 5 years. Finally, with regard to massive hemoptysis, this occurs more often in diseases that affect two lobes or more, while the lower left lobe is less often involved in episodes of massive hemoptysis [6,8,9,11,83,84,85,86].

It has been found that, over time, the rate of clinical success decreases significantly. From the very first month after BAE, the clinical success rate can be significantly reduced due to the resulting rebleeding cases, which can be fatal. For this reason, long-term follow-up is considered useful in order to monitor the course of the patient’s disease. In particular, certain conditions are associated with an increased risk of rebleeding in more than half of patients. Such diseases are bronchiectasis, aspergilloma, and especially lung tumors (primary and metastatic). In these cases, surgical treatment of hemoptysis should be discussed to reduce the risk of rebleeding. Many times, during follow-up, the need for re-interventions can be seen [45,46,47,48,49,50,51,52,53,54,56,57,59,60,61,62,63,64,66,67,68,69,70,71,72,74].

Rebleeding after BAE is the most frequent complication of surgery and determines the prognosis, as it can be fatal. Rebleeding occurs most frequently after the first month to the first year, with a mean day of occurrence of 293 postoperative days. It next most frequently appears in the first month after BAE. Rarely, however, rebleeding can occur after one year of BAE. However, these numbers vary quite a bit and are determined by the existence or not of risk factors. Such factors are diabetes, aspergilloma, and the existence of an arteriovenous shunt. In particular, with regard to aspergilloma, rebleeding can reach 100%, emphasizing the need for regular follow-up and discussion of surgical lobectomy if the patient is deemed cardiorespiratorily fit for such an operation [45,46,47,48,49,50,51,52,53,54,56,57,59,60,61,62,63,64,66,67,68,69,70,71,72,74].

Paresis, a serious complication, was reported twice in the included studies in this systematic review. Mishra et al. [71] documented monoparesis of the right lower extremity after BAE. This paresis was blamed on a small focal hyperintensity of the spinal cord due to a possible embolism of the spinal artery, as confirmed by magnetic resonance imaging. This complication requires physiotherapy that leads to partial-to-complete restoration of the damage. Other serious complications reported include aortic dissection, which occurred in one incident due to a fragile aortic wall. Two cases of cerebral infarction were also reported, one of which was attributed to infarction arising from the vertebral artery. This emphasizes the need for special care when embolizing near the vertebral artery. 

Further investigation needs to be undertaken to find ways to treat massive hemoptysis non-invasively. Also, further investigation is required into how to prevent the complications of BAE, such as rebleeding, which occurs at a very high rate in some cases.

This research has certain limitations. One of them concerns the type of studies that were selected, these being retrospective observational cohort studies and not randomized controlled trials. This is due to the non-existence of other types of studies. Another constraint is the restriction of studies for inclusion to those written in the English language, which created publication bias, as well as the literature search being limited to a narrow period of time. Our systematic review does, however, have several strengths. Firstly, there is the rigorous methodology used according to the PRISMA guidelines and the exploration and reporting of all possible treatments of massive hemoptysis, in addition to only recent studies, i.e., those published in the last six years, having been selected.

## 5. Conclusions

In conclusion, massive hemoptysis is a life-threatening condition and is a frequent symptom of various respiratory diseases, such as COVID-19. Its treatment can be achieved in terms of technical and clinical success with the use of bronchial artery embolization. However, when this method is not indicated, the treatment of massive bleeding is achieved by using other equally effective methods. Complications appear in a large number of patients, the most common of which is chest discomfort. Rebleeding is a major problem, with massive bleeding occurring frequently, which can be fatal. Finally, mortality from massive hemoptysis occurs in a low percentage of patients, which reinforces the effectiveness of the methods of dealing with massive hemoptysis.

## Figures and Tables

**Figure 1 jpm-13-01649-f001:**
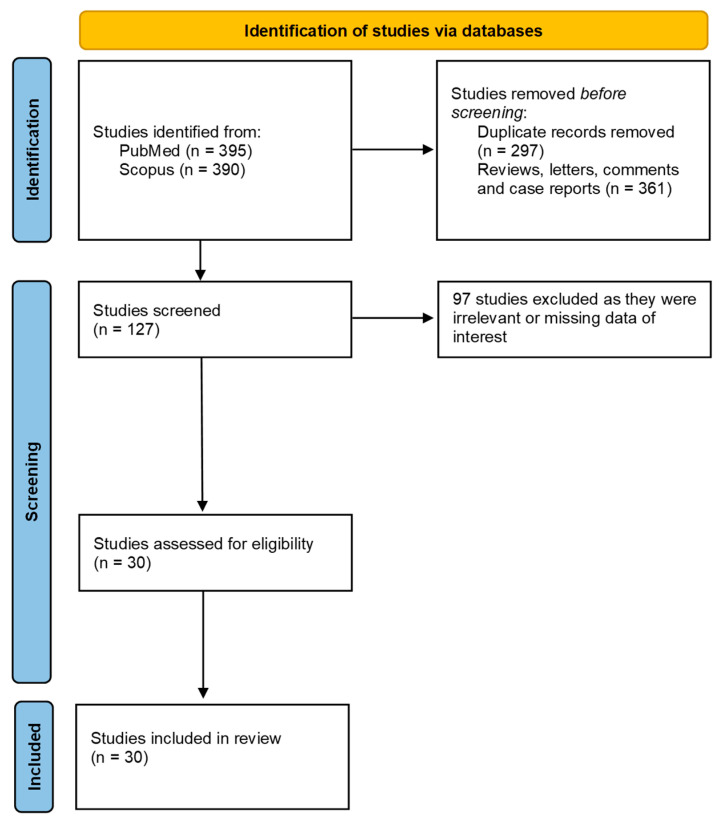
PRISMA 2020 flow chart. This diagram illustrates the process followed for the collection and selection of studies used in our systematic review.

**Figure 2 jpm-13-01649-f002:**
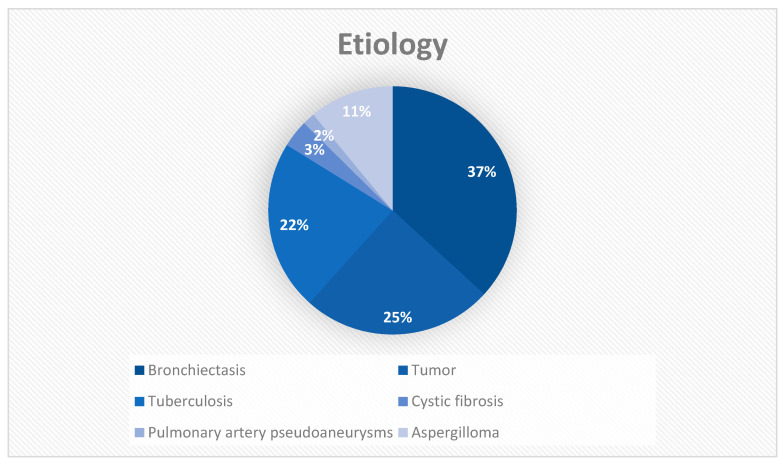
Etiology of massive hemoptysis.

**Figure 3 jpm-13-01649-f003:**
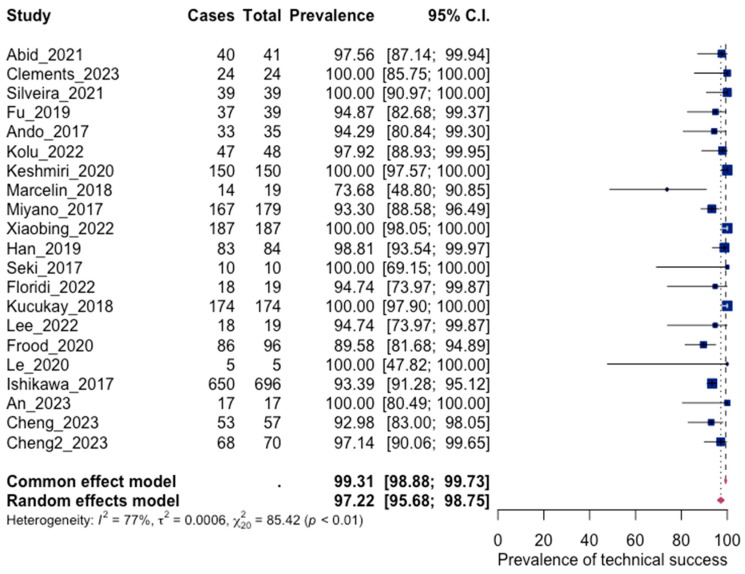
Forest plot for technical success [45,47,49,50,53,54,56,57,59,60,61,62,63,64,66,67,68,70,72,74].

**Figure 4 jpm-13-01649-f004:**
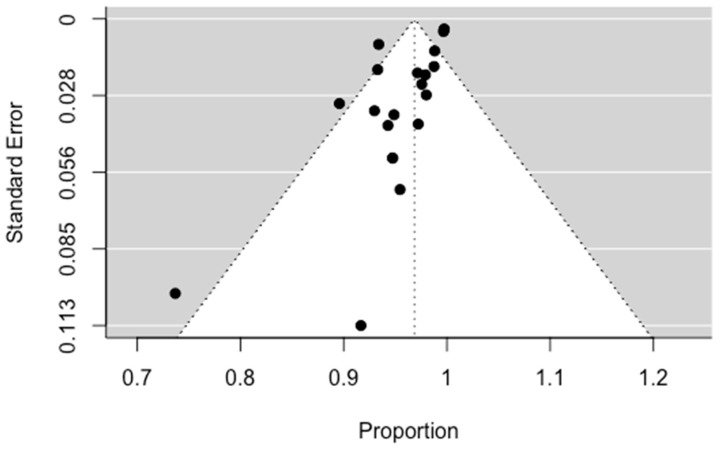
Funnel plot for technical success.

**Figure 5 jpm-13-01649-f005:**
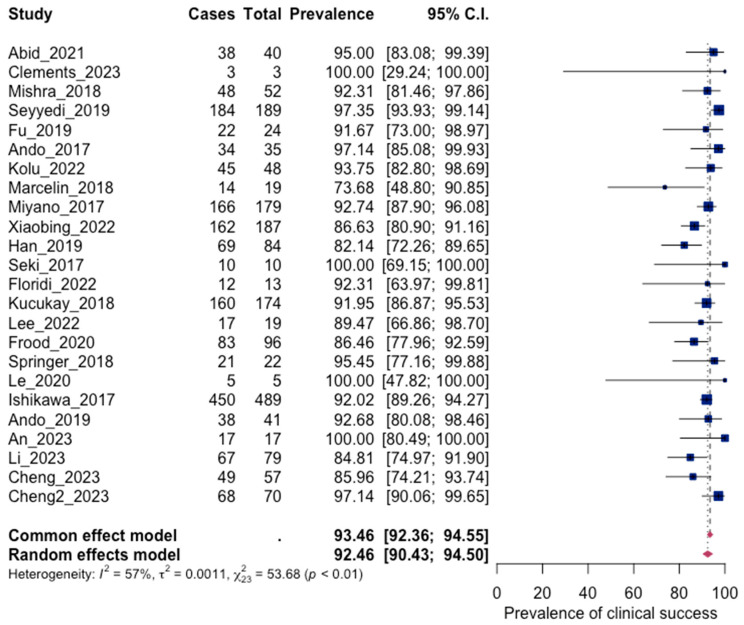
Forest plot for clinical success [45,46,47,48,49,50,51,53,54,56,57,59,60,61,62,63,66,67,68,69,71,72,74].

**Figure 6 jpm-13-01649-f006:**
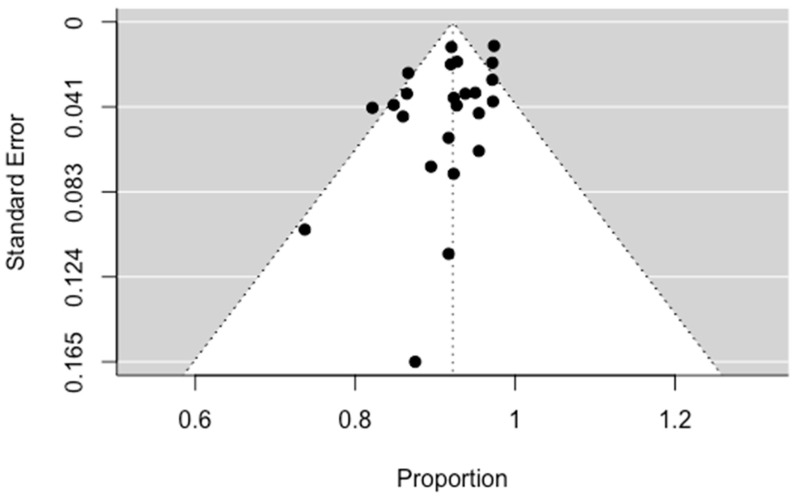
Funnel plot for clinical success.

**Figure 7 jpm-13-01649-f007:**
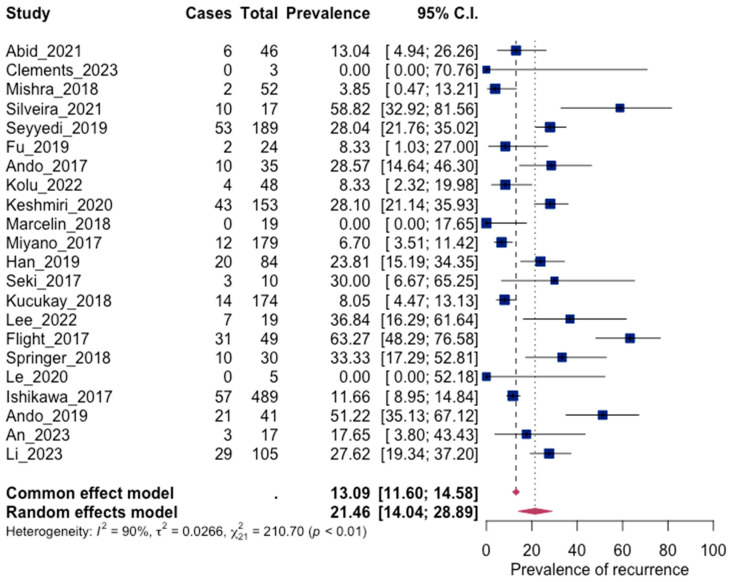
Forest plot for recurrence [46,47,48,49,50,51,52,54,56,59,60,62,63,64,66,67,68,69,70,71,72,74].

**Figure 8 jpm-13-01649-f008:**
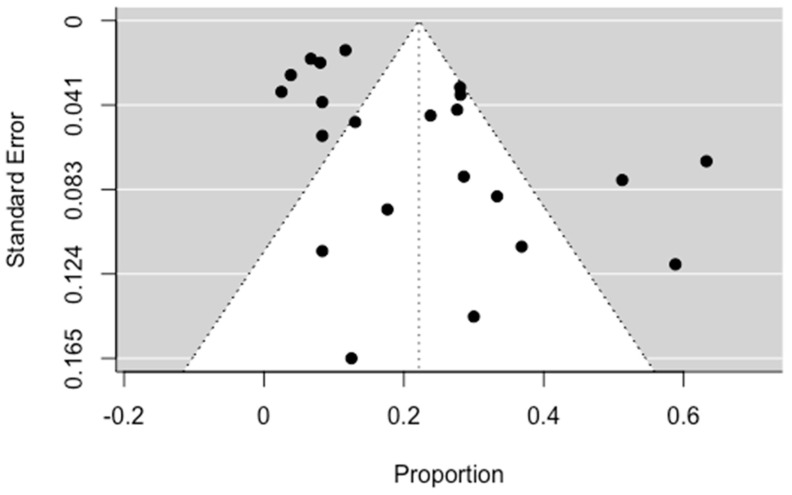
Regression test for funnel plot asymmetry. Significant asymmetry was observed (*p* = 0.0246).

**Figure 9 jpm-13-01649-f009:**
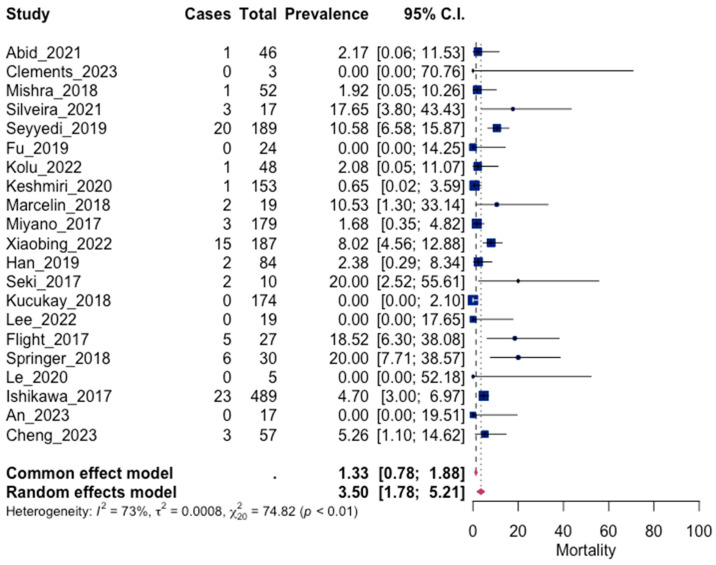
Forest plot for mortality [45,47,49,50,51,52,54,56,59,60,61,62,63,64,66,68,69,70,71,72,74].

**Figure 10 jpm-13-01649-f010:**
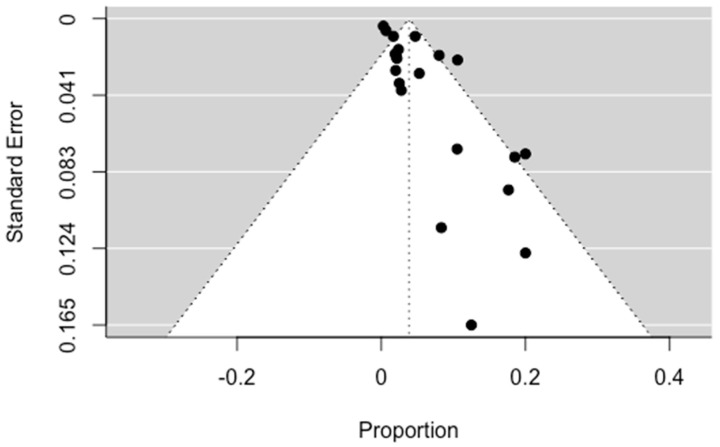
Funnel plot for mortality.

**Table 1 jpm-13-01649-t001:** Characteristics of included studies.

Study ID	Total Subjects (% Male/ Median Age)	Study Type	Management of Hemoptysis	Technical Success Rate	Clinical Success Rate
Abid et al., 2021 [74]	46 (78.2/54.8)	ORCS	BAE	97.5	95
Yang et al., 2022 [73]	15 (80/55.9)	ORCS	Dual-vessel intervention	80	66.7
Clements et al., 2022 [72]	3	ORCS	BAE	100	100
Mishra et al., 2018 [71]	52 (82.6/48)	ORCS	BAE	NM	92
Silveira et al., 2021 [70]	17 (47/25)	ORCS	BAE	100	NM
Seyyedi et al., 2019 [69]	189 (58.3/NM)	ORCS	BAE	NM	97.3
Fu et al., 2019 [68]	24	ORCS	BAE	94.9	91.7
Ando et al., 2017 [67]	35	ORCS	BAE	94.3	97
Kolu et et al., 2022 [66]	48 (68.7/52)	ORCS	BAE	97.9	93.7
Mehta et al., 2020 [65]	12	ORCS	Customized endobronchial silicone blocker	92.3	92.3
Keshmiri et al., 2020 [64]	153 (68/55)	ORCS	BAE	100	NM
Marcelin et al., 2018 [63]	19 (78.9/60.3)	ORCS	BAE	73.7	84.2
Miyano et al., 2017 [62]	179	ORCS	BAE	93.2	92.8
Xiaobing et al., 2022 [61]	187	ORCS	BAE	100	86.6
Han et al., 2019 [60]	84	ORCS	BAE	98.9	82.1
Seki et al., 2017 [59]	10	ORCS	BAE	100	100
Al-Samkari et al., 2019 [58]	21	ORCS	Antifibrinolytic agents	NM	50
Floridi et al., 2022 [57]	13 (NM/NM)	ORCS	BAE	94.7	92.3
Kucukay et al., 2018 [56]	174 (45.9/39.4)	ORCS	BAE	100	91.9
Lal et al., 2021 [55]	27 (NM/ 41.4)	ORCS	Percutaneous transthoracic embolisation	93.1	88.9
Lee et al., 2022 [54]	19	ORCS	BAE	94.7	89.5
Frood et al., 2020 [53]	68 (60.2/53)	ORCS	BAE	90	86.5
Flight et al., 2017 [52]	27	ORCS	BAE		
Springer et al., 2018 [51]	30 (NM/33.5)	ORCS	BAE	NM	95.75
Le-Jun et al., 2020 [50]	5	ORCS	BAE	100	100
Ishikawa et al., 2017 [49]	489 (46.4/69)	ORCS	BAE	93.4	92
Ando et al., 2019 [48]	41	ORCS	BAE	NM	92.7
An et al., 2023 [47]	17	ORCS	BAE	100	100
Li et al., 2023 [46]	105 (64/NM)	ORCS	BAE	NM	84.8
Cheng et al., 2023 [45]	127 (Group E: 47/58, Group G: 53/60)	ORCS	BAE using Embospheres alone and Embospheres with gelfoam particles	92.99 & 97.14	85.96 & 97.14

Abbreviations: ORCS: observational retrospective cohort study, BAE: bronchial artery embolization.

**Table 2 jpm-13-01649-t002:** Etiology of massive hemoptysis in the studies included in this systematic review.

Etiology of Massive Hemoptysis	Number of Incidents
Bronchiectasis	658
Tumor	444
Tuberculosis	397
Cystic fibrosis	64
Pulmonary artery pseudoaneurysms	29
Aspergilloma	196

**Table 3 jpm-13-01649-t003:** Radiological and angiographic findings.

Angiographic Findings	Incidence
Hypertrophy and dilatation	460
Parenchymal hypervascularity	44
Parenchymatous blush	145
Cystic lesion	73
Cavity	37
Aneurysmal lesion	78

**Table 4 jpm-13-01649-t004:** Recurrence, complications, and mortality.

Study ID	Chest Discomfort	Fever	Dysphagia and Throat Discomfort	Paresis	Other Complications	Recurrence	Mortality
Abid et al., 2021 [74]	2	0	0	0	1 (headache)	6/46	1
Yang et al., 2022 [73]	6	0	0	0	12 (ventricular arrhythmia)	2/15	1
Clements et al., 2022 [72]	0	0	0	0	0	0/3	0
Mishra et al., 2018 [71]	5	0	0	1	0	2/52	1
Silveira et al., 2021 [70]	0	0	0	0	0	10/17	3
Seyyedi et al., 2019 [69]	23	0	10	0	6 (subintimal dissection, and pancreatitis)	53/189	20
Fu et al., 2019 [68]	6	2	0	0	1 (vagus reflex)	2/207	0
Ando et al., 2017 [67]	12	0	0	0	0	10/35	
Kolu et et al., 2022 [66]	5	0	0	0	0	4/45	1
Mehta et al., 2020 [65]	0	0	0	0	0	1/13	0
Keshmiri et al., 2020 [64]	0	0	0	0	3 (ischemia, pulmonary infarction, and spinal complications)	43/153	1
Marcelin et al., 2018 [63]	0	0	0	0	0	0/19	2
Miyano et al., 2017 [62]	0	0	0	0	1 (aortic dissection)	12/179	3
Xiaobing et al., 2022 [61]	34	38	0	0	87 (nausea/vomiting, increase in AST/ALT, decrease in platelets, abdominal pain)	NM	15
Han et al., 2019 [60]	0	0	0	0	0	20/84	2
Seki et al., 2017 [59]	1	0	0	0	1 (allergic reaction)	3/10	2/10
Al-Samkari et al., 2019 [58]	NM	NM	NM	NM	NM	NM	NM
Floridi et al., 2022 [57]	NM	NM	NM	NM	NM	NM	NM
Kucukay et al., 2018 [56]	9	NM	NM	NM	NM	14/174	0
Lal et al., 2021 [55]	0	0	0	0	1 (pneumothorax)	1/29	0
Lee et al., 2022 [54]	0	0	0	0	0	7/19	0
Frood et al., 2020 [53]	0	0	0	1	2 (cardio-pulmonary arrest, cerebellar infarction)	NM	NM
Flight et al., 2017 [52]	14	0	0	0	15 (paraesthesia, headache)	31/49	5
Springer et al., 2018 [51]	0	1	0	0	3 (1 haematoma at the site of microcatheter insertion and 2 vomiting)	10/30	6
Le-Jun et al., 2020 [50]	0	0	0	0	0	0	0
Ishikawa et al., 2017 [49]	NM	NM	NM	NM	8 (1 aortic dissection, 2 symptomatic cerebellar infarctions, and 5 mediastinal haematoma cases)	57/489	23
Ando et al., 2019 [48]	NM	NM	NM	NM	NM	21/41	NM
An et al., 2023 [47]	6	5	1	0	0	3/17	0
Li et al., 2023 [46]	8	1	0	0	1 (nausea)	29/64	NM
Cheng et al., 2023 [45]	9	5	0	0	7 (2 vomiting, 2 poor appetite, 2 allergy, 1 transient cortical blindness)	NM	3
TOTAL	140	52	11	2	149	341/1979	89

**Table 5 jpm-13-01649-t005:** Quality assessment of included studies.

Study ID	Selection				Comparability	Outcome			Total	Quality
	Representativeness of the Exposed Cohort	Selection of the Non-Exposed Cohort	Ascertainment of Exposure	Demonstration That Outcome of Interest Was Not Present at Start of Study	Comparability of Cohorts on the Basis of the Design or Analysis Controlled for Confounders	Assessment of Outcome	Follow-Up Long Enough for Outcomes to Occur	Adequacy of Follow-Up of Cohorts		
Yang et al., 2022 [73]	*		*	*	**	*	*	*	8/9	Good
Clements et al., 2022 [72]	*		*	*	**	*	*	*	8/9	Good
Mishra et al., 2018 [71]	*		*	*	**	*	*	*	8/9	Good
Silveira et al., 2021 [70]	*		*	*	**	*	*	*	8/9	Good
Seyyedi et al., 2019 [69]	*		*	*	**	*	*	*	8/9	Good
Fu et al., 2019 [68]	*		*	*	**	*	*	*	8/9	Good
Ando et al., 2017 [67]	*		*	*	**	*	*	*	8/9	Good
Kolu et et al., 2022 [66]	*		*	*	**	*	*	*	8/9	Good
Mehta et al., 2020 [65]	*		*	*	**	*	*	*	8/9	Good
Keshmiri et al., 2020 [64]	*		*	*	**	*	*	*	8/9	Good
Marcelin et al., 2018 [63]	*		*	*	**	*	*	*	8/9	Good
Miyano et al., 2017 [62]	*		*	*	**	*	*	*	8/9	Good
Xiaobing et al., 2022 [61]	*		*	*	**	*	*	*	8/9	Good
Han et al., 2019 [60]	*		*	*	**	*	*	*	8/9	Good
Seki et al., 2017 [59]	*		*	*	**	*	*	*	8/9	Good
Al-Samkari et al., 2019 [58]	*		*	*	**	*	*	*	8/9	Good
Floridi et al., 2022 [57]	*		*	*	**	*	*	*	8/9	Good
Kucukay et al., 2018 [56]	*		*	*	**	*	*	*	8/9	Good
Lal et al., 2021 [55]	*		*	*	**	*	*	*	8/9	Good
Lee et al., 2022 [54]	*		*	*	**	*	*	*	8/9	Good
Frood et al., 2020 [53]	*		*	*	**	*	*	*	8/9	Good
Flight et al., 2017 [52]	*		*	*	**	*	*	*	8/9	Good
Springer et al., 2018 [51]	*		*	*	**	*	*	*	8/9	Good
Le-Jun et al., 2020 [50]	*		*	*	**	*	*	*	8/9	Good
Ishikawa et al., 2017 [49]	*		*	*	**	*	*	*	8/9	Good
Ando et al., 2019 [48]	*		*	*	**	*	*	*	8/9	Good
An et al., 2023 [47]	*		*	*	**	*	*	*	8/9	Good
Li et al., 2023 [46]	*		*	*	**	*	*	*	8/9	Good
Cheng et al., 2023 [45]	*		*	*	**	*	*	*	8/9	Good
Abid et al., 2021 [74]	*		*	*	**	*	*	*	8/9	Good

With *, the star given to each category that excludes the risk of bias is denoted. Each category is evaluated with a blank or a *, except for the Comparability category which is evaluated with a blank or a * or **.

## Data Availability

No new data are created.

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
