# Peer review of "Which Is the Best Way to Treat Massive Hemoptysis? A Systematic Review and Meta-Analysis of Observational Studies"

_jpm, 2023, doi:10.3390/jpm13121649_

Round 1

Reviewer 1 Report

Comments and Suggestions for Authors

I enjoyed reading this paper on the Management of massive Hemoptysis. The strengths of this paper are--great introduction section going over the causes of massive hemoptysis and current management options. It is easy to read and goes into pathophysiology and technical aspects of management. The authors are investigating the success (clinical and technical) of bronchial artery embolization (BAE) in the treatment of Massive hemoptysis. Large sample size(2647 patients). It reconfirms what is the current standard of care, and provides data that BAE is a highly effective way to manage massive hemoptysis. 

Weakness: While it reconfirms the efficacy of the current standard of care, the findings are not Novel. However, despite this limitation given the sample size of this study it should still be published to establish confidence in the efficacy of BAE. Additionally, it is a great in-depth review of the literature on BAE and massive hemoptysis.

Comments on the Quality of English Language

No concerns

Reviewer 2 Report

Comments and Suggestions for Authors

As discussed by the authors, haemoptysis can be a significant problem potentially leading to death. However, many cases are minor and the main goals of management are observation, finding a cause which can be treated, and taking steps to prevent a more severe recurrence. I thus think that the statement " it is an urgent and life-threatening finding" is an exaggeration. Also, how is reference (4) relevant to this statement?

In the Introduction it might be worth pointing out that it is uncommon for bronchial artery bleeding to see the bleeding at the time of the scan, but that it may provide circumstantial evidence about the site and cause of bleeding.

It should be clarified that there are two aspects to the management of haemoptysis. One is how to control and stop bleeding that does not cease spontaneously, and is threatening the life of the patient. This may require rigid bronchoscopy to provide adequate suctioning of the blood, and to intubate the lung that it not bleeding and/or block the airways, such as with a balloon, in the are is bleeding. These measures are designed to control the bleeding and prevent the non-bleeding lung drowning in blood coming from the contralateral lung. More commonly, bleeding stops spontaneously and management is directed at preventing further bleeds. This is the main role for bronchial artery embolization.

Clearer definitions of "technical success" and "clinical success" need to be provided including over what time frame.

In the studies, were the treatments used to stop ongoing bleeding, or prevent recurrent bleeding in the context of massive haemoptysis?

Of course, the opening of the Discussion should explain that there were no randomized controlled trials, so thus the quality of evidence was poor. This does not mean that the studies described in this review are not of interest, but it does mean that definitive statements about the management of massive haemoptysis cannot be made.

Comments on the Quality of English Language

Some minor editing is needed.

Reviewer 3 Report

Comments and Suggestions for Authors

By conducting a systematic review, the authors present the results of the article entitled "the best way to treat massive hemoptysis."

Below are comments that may assist the authors in improving the manuscript:

1-      Selection of articles exclusively in English language has introduced publication bias into the study.

2-      The limitations of this article include its restriction on a narrow time interval for literature search, that it decreases comprehensive search and not including eligible studies.

3-      Considering the inclusion of PICO within the text, it is redundant to include a separate table specifically dedicated to PICO. Therefore, Table 1 should be removed from the study as it duplicates information already presented in the text.

4-      To ensure clarity and focus on the main result, it is recommended that only the funnel plot of the main result remain in the text, and the rest of the funnel plots are included in the supplementary material and removed from the main text of the article.

5-      It has been observed that certain paragraphs within the article contain errors in capitalization, such as the use of capital letters in the middle of sentences. Therefore, it is recommended that the text of the article be thoroughly edited to correct such errors.

Comments on the Quality of English Language

The manuscript exhibits strong academic writing qualities overall. It only requires minor editing.
